# Exploring healthcare-seeking behavior of most vulnerable groups amid the COVID-19 pandemic in the humanitarian context in Cox's Bazar, Bangladesh: Findings from an exploratory qualitative study

**Muhammad Riaz Hossain** [1]*, **Ateeb Ahmad Parray** [1], **Rafia Sultana** [1], **Bachera Aktar** [1,2], **Sabina Faiz Rashid** [1]

1 The Center of Excellence for Gender, Sexual and Reproductive Health and Rights, BRAC James P Grant School of Public Health, BRAC University, Dhaka, Bangladesh, 2 International Public Health, Liverpool School of Tropical Medicine, Liverpool, United Kingdom

* riazhossain.ju101@gmail.com

**Data Availability Statement:** The available data includes illustrative quotes drawn from the

## Abstract

The COVID-19 pandemic has raised new concerns about healthcare service availability, accessibility, and affordability in complex humanitarian settings where heterogeneous populations reside, such as Rohingya refugees in Bangladesh. This study was conducted in ten Rohingya camps and four wards of the adjacent host communities in Cox's Bazar to understand the factors influencing healthcare-seeking behavior of the most vulnerable groups (MVGs) during COVID-19 pandemic. Data were extracted from 48 in-depth interviews (24 in each community) conducted from November 2020 to March 2021 with pregnant and lactating mothers, adolescent boys and girls, persons with disabilities, elderly people, and single female-household heads. This study adopted Andersen's behavioral model of healthcare-seeking for data analysis. Findings suggest that the healthcare-seeking behavior of the participants amid COVID-19 pandemic in the humanitarian context of Cox's Bazar was influenced by several factors ranging from socioeconomic and demographic, existing gender, cultural and social norms, health beliefs, and various institutional factors. Lack of household-level support, reduced number of healthcare providers at health facilities, and movement restrictions at community level hampered the ability of many participants to seek healthcare services in both Rohingya and host communities. Most of the female participants from both communities required permission and money from their male family members to visit healthcare facilities resulting in less access to healthcare. In both communities, the fear of contracting COVID-19 from healthcare facilities disproportionately affected pregnant mothers, elderly people, and persons with disabilities accessing health services. Additionally, the economic uncertainty had a significant impact on the host communities' ability to pay for healthcare costs. These findings have the potential to influence policies and programs that can improve pandemic preparedness and health system resilience in humanitarian contexts.

transcripts, and these are presented within the paper. All qualitative full transcripts are unavailable to the public because they contain personal identifying information about Rohingya and host participants and very sensitive information concerning Rohingya individuals seeking health care outside the camps, which is restricted. Ethical approval for the conduct of the study required that all data, including the locations of Rohingya participant camps, be de-identified, according to the data sharing policy of the Institutional Review Board (IRB) of BRAC University – an independent research ethics committee under BRAC University (https://bracpgsph.org/research-irb). For data access, qualified researchers may request de-identified data by emailing IRB of BRAC University at irb-jpgsph@bracu.ac.bd. The IRB is located at BRAC James P Grant School of Public Health, BRAC University 6th Floor, Medona Tower, 28 Mohakhali Commercial Area, Bir Uttom A K Khandakar Road, Dhaka-1213, Bangladesh.

**Funding:** This study was supported by the BRAC James P Grant School of Public Health, BRAC University. No additional external funding was received for this study. The funders had no role in study design, data collection and analysis, the decision to publish, or preparation of the manuscript.

**Competing interests:** The authors have declared that no competing interests exist.

## Introduction

Rohingya are one of the world's most marginalized and oppressed groups, consisting primarily of the Muslim ethnic minority in Rakhine State, western Myanmar, who are not recognized as citizens by the government [1]. In 2017, over 750 000 Rohingya people were forced to flee to Cox's Bazar, Bangladesh after being exposed to massive human rights violations committed by the Myanmar army [2]. An estimated one million Rohingya refugees live in 34 camps surrounded by a host community of around 335,900 people [3] one of Bangladesh's poorest southern districts. COVID-19 pandemic disproportionately affects certain populations as a result of underlying wealth disparities, social discet alrimination, and social exclusion [4] particularly the poor and most vulnerable populations living in humanitarian setting in Cox's Bazar [5]. Their ability to meet its current health needs is limited because of uneven access to health services, a lack of healthcare workers, and inadequate hospital facilities [6]. Therefore the task of identifying and treating a large number of COVID 19 patients put enormous strain on healthcare systems and had a significant impact on the healthcare workforce [7].

The Government of Bangladesh and the Office of the Refugee Relief and Repatriation Commissioner (RRRC) imposed movement restrictions on March 26, 2020, and other mitigation measures in camps and host areas, impairing health, Water, Sanitation and Hygiene (WASH), and other relevant services [8, 9]. During the lockdown, regular treatment for the elderly, disabled, and women was difficult in both communities due to transportation issues, a lack of medical facilities, financial constraints, and the fear of infection from other COVID 19 patients [10]. The inability of health workers to provide door-to-door services in the community has a negative influence on maternal and child health, as well as sexual and reproductive health care for married adolescents, pregnant women, and women [10]. As well as transportation is restricted, people with COVID 19 symptoms have less access to testing centres. This overall services for other type of sickness or disease, and resulted admissions to health facilities decreased at the time [11]. In COVID-19 situations, there is a shortage of empirical evidence examining the access barriers encountered by the most vulnerable peoples.

In this paper we elicited evidence on factors influencing the healthcare-seeking behavior of MVGs during the COVID-19 pandemic in the humanitarian context of Cox's Bazar involving both the Rohingya refugees and their adjacent host communities. Identification of these determinants will enable national and international decision-makers in developing a balance between resource allocation and service planning in order to improve healthcare system during a pandemic, particularly for the most vulnerable population in terms of access to healthcare services.

## Materials and methods

### Study design

This paper presents findings from the exploratory qualitative part of a mixed-method research conducted in ten Rohingya camps and four wards of the adjacent host communities in the Ukhiya sub-district of Cox's Bazar, Bangladesh. This focused on awareness, preparedness, and impact of the COVID-19 pandemic among the MVGs in both the Rohingya and host communities [12], see below for more details.

### Setting

This research was conducted in the Ukhiya sub-district of Cox's Bazar, Bangladesh, where approximately 9,00,000 Rohingya in 34 camps and 335,900 host communities live adjacent to each other predominantly rural areas of Ukhiya and Teknaf subdistricts [13].

They are one of the poorest communities in the country, with a poverty rate of 33% [13]. They are integrated with the formal service delivery model of the GoB [References], which includes health care provision, primarily by the government-owned facilities, [14]. Additionally, some non-governmental organizations (NGOs) complement these services.

### Eligibility and participant recruitment

This study utilized a subset of the larger participatory action research project [12] in which the groups deemed most vulnerable (MVGs) during the COVID-19 pandemic in the studied contexts were identified based on a criterion developed through exhaustive literature review, expert consultation workshop with government and non-government humanitarian stakeholders working in Cox's Bazar, and validated with community people through field visits to both communities

### Conceptual framework

We adopted Andersen's behavior model as guiding frameworks for our exploration of factors influencing healthcare-seeking behavior in the Rohingya and host community contexts (Fig 1) [15–18]. The model included social and economic variables [19] in combination of both contextual and individual factors that affect health system, healthcare service providers and the community people's healthcare seeking behaviour, and experiences with service providers [20]. During the data analysis and conceptualization of the model, the source of health information emerged as a new factor and was incorporated into our conceptual framework. Fig 1 depicts our conceptual framework (HSB-MVG conceptual framework) which analyses healthcare-seeking behavior across four domains: socioeconomic and demographic factors, individual factors, health belief factors, and institutional factors.

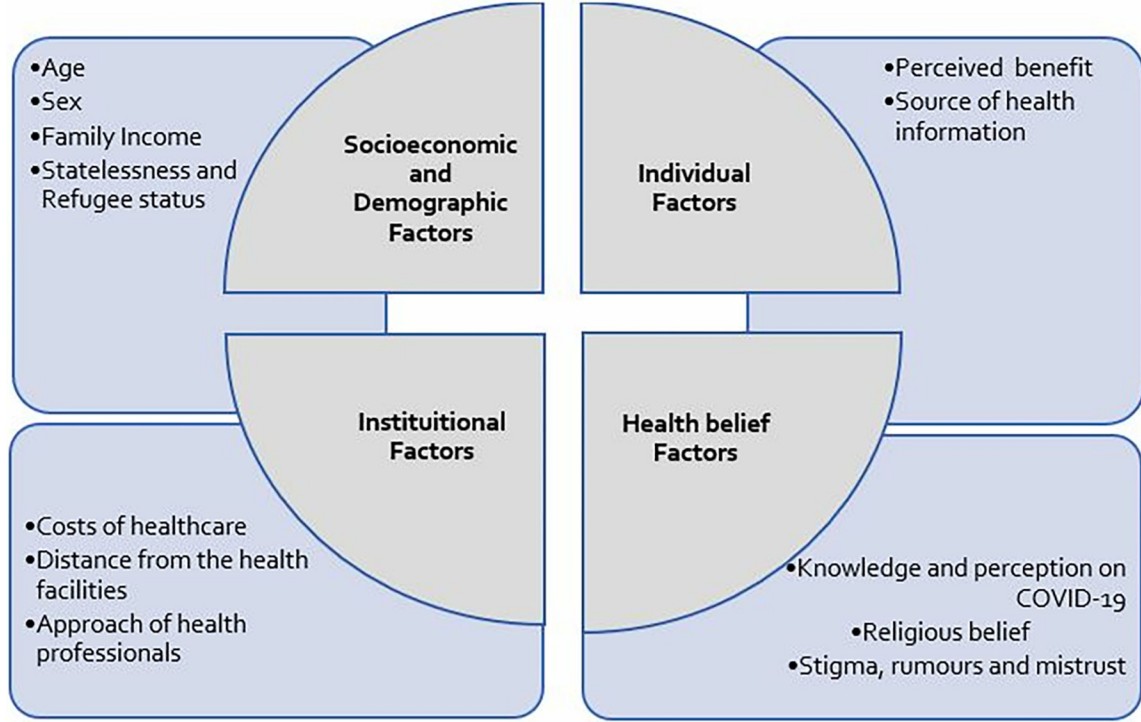

**Fig 1. Conceptual framework of the study- The HSB-MVG conceptual framework.**

## Sampling and data collection

Data were collected from men and women, aged 11–65+ years, from the Rohingya and host communities, using in-depth interviews conducted between Nov 2020 and Mar 2021. In both communities, a total of 48 in-depth interviews were done, 24 in each; 12 with men and 12 with women. Participants were chosen at random based on the pre-identified variables for MVGs from the literature review, formal healthcare providers, program implementors and community influential. During data collection, in order to eliminate biases and reach a degree of saturation, the researchers approach participants from every block of the selected camps and regions of the local community. Table 1 shows data collection method, study population, and sampling. Six trained qualitative researchers (4 men, 2 women) used a pre-tested interview guideline to conduct interviews. 12 experienced and trained local enumerators (4 males and 8 females) who knew the local context and language of both communities acted as interpreters during IDIs. 40 to 90 minutes per interview. Bangla (the host community's language] and Ruáingga were used [the language of the Rohingya community). Before the interview, each participant gave verbal consent (or assent from their parent/guardian). All participant interviews were audio-recorded, and field notes were taken. BRAC JPGSPH's data management policy for the research study was followed.

## Data analysis

After data collection, local enumerators transcribed the data verbatim in Bangla within 24 hours to reduce data loss. Professional translators then translated the data into English. The researchers next checked the data's veracity. The conceptual framework guided inductive and deductive coding using AtlasTi software. Two senior researchers (SFR and BA) assessed a preliminary data display matrix that summarized the primary themes and subthemes. Discrepancies were discussed, resolved, and reviewed to obtain consensus. The data were combined into a final data display matrix before framework analysis. Open-access referencing was used (Mendeley Desktop).

## Ethics considerations

The BRAC James P Grant School of Public Health Institutional Review Board (IRB) [21] approved the study (IRB-6 November'20–057). Due to the political sensitivity of the Rohingya situation in Bangladesh, Rohingya refugees were reluctant to offer written consent but eager to participate in the study. Locals refuse to give written authorization to participate. MVGs have trouble reading and writing. Due to the illegal drug trade and police activities, there was a lot

**Table 1. Data collection method, study population, and sampling.**

| Data Collection Method | Type of MVGs | Sample Size | | | | Age |
|---|---|---|---|---|---|---|
| | | Host | | Rohingya | | Years |
| | | Male | Female | Male | Female | |
| In-depth Interview | Single Female Household Heads | - | 3 | - | 3 | 21–45 |
| | Pregnant and Lactating Mothers | - | 3 | - | 3 | 16–35 |
| | Adolescents | 3 | 3 | 3 | 3 | 11–17 |
| | Persons with Disabilities | 3 | 3 | 3 | 3 | 16–45 |
| | Elderly | 3 | 3 | 3 | 3 | 65+ |
| Total | | 9 | 15 | 9 | 15 | |
| | | 24 | | 24 | | |
| | | 48 | | | | |

of mistrust in the neighborhood; as a result, it was stressful for community members to sign paperwork for an outsider. After reading the study's information page and explaining its objective, participants gave verbal agreement. Given the setting, the authors' IRB approved oral consent. Adolescents gave verbal assent, and their guardians or parent gave approvals. Audio transcripts included oral consents. Data collection, storage, and analysis respected participants' privacy and confidentiality. Each interview was held in a private location (usually the participant's home). Only minimal personal information, such as age, gender, occupation, and education, was obtained under pseudonyms. All information, including Bangla and English transcripts, was held in a safe locker at the researcher's home institute. The research team implemented COVID-19 safety precautions during fieldwork to prevent coronavirus transmission to community members, research participants, and researchers. As a thank-you for participating, all participants received hygiene packets (facial masks, soap, toothpaste, toothbrush shampoo).

## Results

### Profile of the participants

Among the 48 participants, three single female household heads aged 21–45 years, three pregnant and lactating mothers aged 16–35 years, three male and three female adolescents aged 11–17 years, three male and three female persons with disabilities aged 16–45 years, three male and three female elderly population aged 65+ from each community. The Rohingya adolescent participants, both boys and girls were not studying during data collection, while the host adolescents were all students. In both groups of participants, only two participants from the host community were involved in income generation activities, such as small businesses and private jobs.

### 1) Socioeconomic and demographic factors

According to our analysis, socioeconomic and demographic factors like age, sex, family income, and ethnicity were found to be some of the key factors that determined the healthcare-seeking behavior of our participants.

**Age, sex, and family income.** The findings revealed that in both Rohingya and host communities, women, adolescent girls and elderly people were dependent on their family members for accessing the healthcare facilities. This was mainly due to their limited mobility driven by their weak physical health in the case of older adult participants, and strong socio-cultural, religious, and gender norms in the case of women and adolescent girls, and financial dependency for all three groups.

During the interviews (April 2020—June 2020), many of elderly participants stressed that due to their weak physical condition, they required their family members to accompany them to their routine healthcare facilities but, due to the fear of contracting COVID-19 in the health facilities, their caregivers refused to accompany them. As elderly participants shared, One 70 years old male Rohingya participant shared,

"*When they asked us not to go outside [lockdown], my grandson stopped taking me to see the doctor. He was afraid that he would get sick of the Coronavirus [. . .]. I cannot argue with him because he looks after me*" (70 years old male Rohingya). "*I asked my son to take me to hospital many times. He said he would not go anywhere during the lockdown. But the lockdown did not seem to end*" (68 years old female Rohingya).

Similar incidences were also shared by our younger participants as well, mostly the Rohingya adolescents. However, unlike the elderly Rohingya participants, their dependency on family members was driven mainly by two factors; (a) strong socio-cultural, religious, and gender norms that restricted their mobility to venture outside without their family members, and (b) their financial dependence on their families. As mentioned by Rahima [pseudonym], a 14-year-old Rohingya girl,

"*I need medicine for my headache. Usually, I go to the hospital with my brother and get it from nurse Apa. But when the lockdown was enforced, my brother refused to come with me. My father does not allow me to go outside alone. I had to suffer for weeks because of my headache.*" (14-year-old girl, unemployed, Rohingya).

Most of the Rohingya girl participants mentioned that they faced difficulties while accessing healthcare facilities that their male siblings did not face as the society did not impose rigid norms on adolescent boys. As a result, many of them admitted seeking help from their male family members whenever they faced any health-related issues or when adolescent girls and women needed the menstrual kit. As was in the case of Shahida (pseudonym), a 16-year-old Rohingya girl, who mentioned

"*I had to explain my problems to my father or brother who would go to the drug store and get medicine for me [...] You can understand, how is it possible for a girl like me to explain every problem to my father and brother. Not every concern can be shared with men.*" (16-year-old girl, Unemployed, Rohingya).

In the Host communities' context, similar to the Rohingya camps, most elderly participants, regardless of gender, mentioned that their family members refused to accompany them to the health facilities due to fear of contracting COVID-19. As a result, many opted to seek healthcare treatment from the informal providers, "*Vaidyas,*" (a practitioner of Ayurvedic medicine or folk medicine).

In contrast, many adolescent girls, including young lactating mothers in the host communities, mentioned that their family income had reduced due to the COVID-19 lockdown. As a result, their family members were reluctant to accompany them to the hospitals as that would incur transportation costs. Additionally, the strict socio-cultural, religious, and gender norms restricted females from venturing outside their homes independently. Since they did not earn, they depended on their family members for day-to-day expenses. Therefore, many female adolescent and women participants mentioned seeking health-related advice and services from informal providers in the neighborhood to avoid transportation costs.

Thus, informal providers were perceived as the most viable and accessible source of healthcare, given the existing constraints, now exacerbated by COVID-19. As mentioned by one 17-year-old adolescent girl,

"*In the lockdown, my father refused to go with me to the hospital. I wanted to go alone, but [...] my family will never allow me to go that far alone [...] So, I always take medicine from a nearby pharmacy [local drug store]. [...] I tell the Bhaiya there (Salesperson) what my problem is, and he gives me medicine [...] It costs some money, but we save the transportation cost*". (17-year-old adolescent girl, unemployed, Host).

Almost all of our participants in the single female household heads' group from the host communities unanimously agreed that loss of income during the COVID-19 lockdown had

impacted their personal lives, leaving them struggling to survive. Some of them mentioned that they had to skip meals so that their children could eat well. As a result, health was at the bottom of the hierarchy of their priorities during the pandemic. In the words of one single mother,

*"Since the lockdown was announced, I did not seek any health services even when I was sick. If I go to any hospital, I need money to cover my transportation cost, which was high in the lockdown period. As I could not manage meals for my family, I've seen my kids crying for food and forgot about my sickness"* (43 years old single female-headed HH, Unemployed, Host).

## Statelessness and refugee status

For the Rohingya community, having or not having a formal refugee status was found as an important factor influencing their healthcare-seeking behavior.

One-fifth of our Rohingya participants lived in "the registered camp" in Cox's Bazar. These participants were forcibly displaced in the early 1970s and had "refugee status," which only residents of the registered camp (out of 34). All of these individuals had easy access to local health facilities. This was an advantage as it provided wider alternate health care facility options in the host community for them during the initial stages of COVID-19 lockdown. However, for new Rohingya participants (migrated during the recent influx in August 2017) residing in other new makeshift camps and not recognized as registered refugees by the Government of Bangladesh, designated health centres inside or close to their camps were the only option for receiving formal healthcare services. They were not allowed to go to any health centres in the host community or even health centres in other camps without permission from the respective camp authority. Although health services were free in the designated health centres and easily accessible to the Rohingya refugees, most of the Rohingya participants across all groups mentioned that during the COVID-19 pandemic, the medical staff was frequently unavailable in those facilities. One 65 years old male Rohingya participant with physical disability shared his experience,

*"During the lockdown, I had severe chest pain. My son and one neighbor took me to many health facilities in the camp, but no doctors were available. [. . .] So, we were able to go outside the camp [in the host communities] to one hospital [. . .]."*.

## 2 Health belief factors

According to our analysis, factors like knowledge and attitude regarding the COVID-19 were some of the key factors that determined the health-seeking behavior of our participants during the COVID-19 lockdown.

**Knowledge and perception on COVID-19.** The data revealed that participants had varying levels of knowledge and, perceptions regarding COVID-1. Their understanding of the disease and believe influenced their desire to seek care, decisions and selection of providers during the COVID-19 lockdown.

Many of the adult Rohingya participants, mostly elderly, PWDs and some single female household heads perceived COVID-19 as a "*Gujob*" (rumor), some said, "*Allah r Azab*" (wrath of God), while a few mentioned, "*Allah r porikkha*" (God's test). Driven by those religious believes, most of them found reluctant to follow COVID-19 safety measures recommended by the health providers and camp authorities.

Similarly, many adult participants from the host community, mostly single women, pregnant and lactating women, and some PWDs, attributed COVID-19's origin and causes to

religious punishment same as Rohingyas. At the same time, most of the Hindu women described COVID-19 as "*Mondho Bhaggo*" (Bad Fate). Some elderly participants in the host community also believed that Muslims won't be infected by COVID-19 and being a Muslim they were safe from the harmful effects of COVID-19. In both cases, the fatalistic attribution of COVID-19 to religious beliefs made many participants perceive themselves as safe from COVID-19, as it was inevitable that they would be infected if they were 'sinners.' As mentioned by a 38-year-old single mother,

> "*COVID-19 is the result of Mondho Bhaggo (Bad Fate). If your actions are good, you do not need to worry. I am a destitute woman. I have never wronged anyone. How will COVID-19 kill me*?" (38-year-old, Single Mother and Homemaker, Host).

**Stigma, rumours and mistrust.** Stigma and rumours against the treatment and management of COVID-19 in the Rohingya camps during the lockdown prevented many participants from going to the health facilities. a 22-year-old girl was afraid that she would also be seized and thus stopped visiting the health facilities during the lockdown. In her own words,

> "*My grandmother told me that if you have cough these days, they [camp officials] will not let you come back from the health posts. They will take you to center [isolation center] where God knows what they will do to you*" (22-year-old, PWD, Rohingya).

However, after the lockdown was enforced, in the case of boys, the rumours, fears, and stigma around healthcare facilities were noted as the primary reason for seeking healthcare from alternate sources. A 16-year-old boy scared and sought treatment from a local pharmacy close to his house but not the health facility. In his own words,

> "*During the lockdown, when I got sick, I was afraid to go to the health post because I thought even if I do not have COVID-19, they [hospital staff] will [either] detain me as suspected COVID-19 patient, or I will catch COVID-19 from them anyway.*" (16-year-old, Day Labourer, Rohingya).

Being refugees and already persecuted, there was inherent fear and mistrust of the administration's agenda and most remained anxious that they would be taken away into 'isolation' and would never see their family members again.

Similar stigma was reported by host participants as well. The host community participants also had diverse opinion towards COVID-19 driven by socio-cultural and mistrust. Especially pregnant women, elderly women, and adult participants were unwilling to visit any health facility despite having COVID 19 symptoms due to fear and a lack of adequate information about COVID 19 isolation and social stigma associated with the infection.

## 3) Individual factors

The individual factors influencing the healthcare seeking behaviour cited were perceived benefit and source of health information which also determined the healthcare seeking behavior of the participants.

**Perceived benefit.** Our data suggest a link between participants' perceived benefit and their health-seeking behavior amid the pandemic. All groups, despite different reasons, mainly had inclined towards informal providers such as traditional healers and/or the medication prescribed by the local drug store salesperson "*Oshud Dokaaner Bhaiya*," often perceived as "*Village Doctors*,", which were more accessible to them during the COVID-19 lockdown, and they

felt comfortable with them. Burmese doctors (informal providers from their country) played a huge role in the lives of these communities as mentioned by our participants. Also the Imam [religious leader] placed a role for general health treatment. People visit them and collect "Panipora" and "Piaj pora" (Recite religious verse and blow in the water) for cold related symptoms, mild fever, Jaundice and stomach ache.

Similarly, one pregnant woman mentioned that a "*Vaidya*" [Traditional healer; informal provider] had suggested her drink "*aadha cha*" (Ginger tea) every day as a preventive measure for COVID-19.

**Sources of health information.**   The study found that the preference and trust on the source of health information influence people's healthcare seeking behaviour in the both study communities. At this stretch, the individual COVID-19 health-seeking behavior of participants in both communities had been influenced by awareness information from trusted sources for example religious, political, and social leaders, Community health workers, and digital media. However, our data suggest substantial gender differences between the participants' social networks as sources of health information which also help them to choose health care providers.

In the Rohingya context, male participants perceived their religious leaders as the most reliable source of health information during the COVID-19. However, the female participants expressed their vote towards the community health workers and government officials as the most reliable sources of health information during the COVID-19. Like Rohingya participants in the host community group, most of the male participants expressed an inclination towards religious leaders and local government officials as the most reliable sources of health information. In contrast, the female participants preferred door-to-door meetings with NGO-based community health workers because that allowed for more private space to talk about their day-to-day issues.

## 4) Institutional factors

Our participants' healthcare-seeking behavior was found to be influenced by characteristics such as the price of health services, the distance from health facilities, the behavior of health professionals, and their level of satisfaction with health services.

**Costs of healthcare.**   This study found that many MVG participants could not avail themselves of their essential health services during the lockdown due to their inability to afford medicine and pay diagnostic fees. Most of the MVG participants from the Rohingya and host communities mentioned that they lost their only earning source during the COVID-19 lockdown. This directly impacted their family income, which led to economic uncertainty within their lives affecting their ability to spend money on healthcare services, including doctor's fees, diagnostic tests, and essential medicine.

According to the Rohingya participants, they could avail required medicine free of cost from some of the health posts before the lockdown. However, many MVG participants mentioned that those health posts were closed during the first month of the lockdown, prompting them to go to the local drug shops to buy the medicines. Hence, many elderly and PWD participants who constantly needed regular treatment could not afford their medication. As one elderly female participant narrated,

*"I could not survive a day without my medicine. My knee pain did not let me sleep at night. Before the lockdown, I did not require any money for my regular medicine. But the hospital where I used to go to collect this medicine was closed during the first month of the lockdown. Thereby, I had to pay 400 BDT to get my medicine from a local drug store. Hence, I could not continue this medicine for that month. I was left with no other choice but either to beg from*

*my neighbor to collect money for my medicine or discontinue this for one month.*" (68-year-old female, Unemployed, Rohingya)

This situation was comparatively worse in the host community than in Rohingya camps. As per most of participants, a few health facilities were open during the lockdown period. Those facilities did not provide any free services, and hence people who lost their job and monthly earnings often could not manage money for their treatment. As a result, some MVG participants skipped their treatment during the pandemic or purchased their medicine from the local drug stores and other informal providers, where medicines could be purchased at time on credit.

**Distance from the health facilities.**   Distance from the health facilities was one of the significant barriers to accessing healthcare facilities for all the MVGs in both communities; particularly the pregnant and lactating mother, PDW, and elderly participants. Our data suggest that the lack of support personnel and limited transport facilities hindered them from visiting the health facilities amid the pandemic.

In the Rohingya Camps, multiple health facilities remained open during the lockdown. However, the majority of the PWD and elderly participants shared their struggle in terms of availing the health services due to the distance of the health facilities and the lack of accompanying persons. Not all the health facilities were easily accessible for the community people as for the hilly terrain and most of the time they had to walk 15-minutes to an hour to reach the health facility. Their caregivers did not want to take them to the hospitals as they thought the hospitals could be the areas from where they could be infected. Moreover, PWD, elderly, or females both single and pregnant could not go to those health facilities alone since they were not near their homes. As a result, these groups of people often chose to avoid visiting those facilities during the lockdown. As one elderly female participant narrated,

*"I don't have a son or a daughter. if I fall sick there's no one to look after me. I am helpless. There's even no one who would take me to the doctor." (65-year-old female, Unemployed. Rohingya)*

Similar experiences were shared by many participants from the host community as well. According to them, they could not visit the hospitals because of the absence of caregivers and the unavailability of transportation. Due to this reason, sometimes the external caregivers (mostly their neighbors) demanded extra money to accompany them to the hospitals. As one PWD participant shared,

"*Visiting a doctor during the lockdown was not easy for me. Since I have difficulties walking, I always need assistance when I go out. . . During the lockdown, it was quite difficult to find caregivers because they did not want to go to the hospitals. . .they thought they would get infected with the virus from the hospitals*" (30-year-old male, Unemployed, Host). Therefore, they opted to visit informal providers who lived nearby and were thus more accessible and cost-effective.

**Approach of health professionals.**   In the studied context, one of the primary reasons for the unwillingness to avail healthcare services from the health facilities was the health professionals' approaches towards a patient.

In the Rohingya community, though many of the health facilities were open amid the pandemic, the utilization of health facilities was less due to the reported unsatisfactory behavior of the healthcare staff. The majority of the participants, including the pregnant and lactating

mothers, PWD, and elderly, reported their dissatisfaction regarding the doctors and medical staff. According to them, the doctors did not perform appropriate check-ups since the pandemic started. Many of them pointed out that before the pandemic, doctors performed comprehensive check-ups using multiple instruments, and physical touch. However, they experienced a changed behavior during the pandemic as the medical staff specifically the doctors sat very far from them, did not touch them, and expressed hurry while listening to their problems. One elderly male participant shared, *"The doctor did not even come near me or check me properly. . . I am just upset by their behavior. Why would they do that?"* (64-year-old male, Unemployed, Rohingya). "

A similar experience was shared by one of the PWD participants as well. As he narrated,

*"We had to stand 4 feet away from the doctors. How could a doctor check me up properly if he did not come close to me? I am not sure about the quality of the treatment given during the pandemic"* (34-year-old male, Unemployed, Rohingya).

In addition, a number of participants expressed discontent with the quantity and variety of medicines they received from the health centre. Respondents also reported that doctors recommended paracetamol-based medications for the majority of their disease treatments without conducting a thorough assessment. Furthermore the facility did not give all of the prescribed medications. Due to this reason, patients were not recovering as early as the pre-pandemic period. Therefore, many opted to visit the informal providers since they listened to more time and care. Hence, they prescribed them appropriate medicine as believed by the participants. As one pregnant woman said, *"We visited the health care facilities for treatment, but health facilities provided us with less medicine during this pandemic. They only provided paracetamol which we bought from outside. So, what is the point of going to the health facilities? Rather we could go to the pharmacy where doctors (attending person) would listen to our problems with more time and could give us the proper medicine"* (23-year-old female, Homemaker, Rohingya).

In the host community, most participants shared experiences. Before the pandemic, most of the host people went to seek healthcare services from Government-owned health facilities. Since the medical staff was not compassionate towards the patients, some participants did not go to those hospitals anymore, prompting them to go to the informal health care provider (*Vaidya*) instead. As one PWD participant shared,

*"Getting quality treatment was so challenging during the pandemic. Doctors maintained the distance strictly for everything. If the condition was too serious, they wore gloves to check the patient and prescribe medicine. . .I thought, what is this disease that they won't even check up on properly? ' I had doubts whether the prescribed medicines were right or not. If the health service quality was that bad, why would we go to the healthcare facilities? We can rather go to the nearby 'Vaidya,' who would treat us even better than them"* (28-year-old female, Homemaker, Host)

One single female household head shared a similar experience. She said,

*"The doctors examined their patients from a long-distance during the pandemic. Even if the patient were in critical condition, the doctors would perform a check-up from a distance. The patient might die because of this, right?* (55-year-old female, Homemaker, Host). Additionally, 'talking while wearing a mask' was identified as another problem; the doctors could not hear the patients clearly, and they could not understand what doctors were saying. This also refrained them from visiting the health care facilities during the pandemic

## Discussions

This study was conducted with the aim to explore the factors influencing the healthcare-seeking behavior of MVGs in the selected Rohingya camps and adjacent host communities of Cox's Bazar amid the COVID-19 pandemic. It provides specific information on the pandemic scenario, socioeconomic constraints, geographical conditions, and individual aspects that make it more challenging for MVGs to access healthcare treatment in a humanitarian context.

Mobility restriction was identified as a barrier for PWDs, older adults, and pregnant women because of their age and physical condition. One study, done in a similar context demonstrated that individuals with physical disabilities were dependent on others to access health care facilities in Bangladesh [22]. However, our study found that a group's sex and age are also significant factors in terms of seeking healthcare services. Due to socio-cultural, religious, financial dependency, and gender norms, adolescent girls in our study groups from both Rohingya and host communities expressed the need for an accompanying person and permission from the family heads to visit the health facilities.

Evidence suggests that, at the household level, girls faced increased responsibility for caregiving and domestic chores during the COVID-19 pandemic globally [23, 24]. If resources were scarce, families often prioritize food over healthcare as in the case of single female households. In many places, restrictive gender norms also limited girls' mobility and voice, thereby reducing their access to basic healthcare services [23, 24]. Rigid gender norms, coupled with movement restrictions in an already humanitarian fragile context, have also increased the risk for girls experiencing multiple forms of violence and other harmful practices such as child, early, and forced marriage [25].

The COVID-19 pandemic demonstrated the effects of a contemporary worldwide catastrophe on the healthcare system, particularly in a humanitarian context like Yemen [26]. Participants from our study also mentioned the consistent unavailability of essential medicine, lack of healthcare providers, and lack of sympathetic behavior from healthcare providers during their visits which created a negative impact on them towards formal healthcare services. During the COVID-19 lockdown period, it was challenging for all MVGs to reach the healthcare facilities because of the movement restrictions from the government, unavailability of support personnel, fear of contracting the virus from the health facility, and in some cases, higher travel costs as well. The healthcare system seems to have shifted responsibility for making ethical and technical decisions to individuals by letting patients choose the level of urgency of the care they require, changing the dynamics of how people use healthcare services [27]. However, when they reached the facilities and had to return without treatment due to the absence of healthcare providers or/and their ill behavior, it created an unsatisfactory impression towards formal healthcare providers across the communities. This type of experience has far-reaching consequences for a larger community since poor quality healthcare services can extend beyond preventing health improvement; they can also increase the monetary burden of illness and healthcare expenditures [28].

In the world stage, countries with insufficient health systems like Bangladesh are overburdened by pandemic breakouts when health services are desperately required [29]. Evidence suggests that during the first year of the COVID-19 pandemic, many healthcare workers experienced additional stress, psychological problems, lack of personal protective equipment (PPE), were excluded or stigmatized, lack of sufficient incentives, absence of coordination, and had management issues [29, 30], which may have affected their attendance and behavior. Furthermore, studies showed that frontline healthcare workers in humanitarian contexts are particularly vulnerable to COVID-19 [31] which also added to the stigma and contributed to the absenteeism of healthcare workers during the initial COVID-19 lockdown.

Informal providers were perceived as the most viable and accessible source of healthcare during the COVID-19 lockdown period in both communities. Ease-of-access was identified as an important factor that led the MVGs to seek treatment from informal healthcare providers such as local drug store salespeople, Vaidyas, Burmese doctors, and Kobiraj. In any humanitarian context, the transition of healthcare-seeking behavior of populations from formal to informal may prove detrimental [32] and indicates fragility in the existing healthcare system.

Overall, most of our participants from both groups reported stigma as the main reason for not visiting health facilities fuelled mainly by two beliefs: religion and fear. COVID-19 has been a locus of religious, social, and geopolitical stigma and evidence suggests that the healthcare-seeking behavior of many MVGs has been influenced by this. Our findings synchronize with the global picture of COVID-19 healthcare-seeking behavior during the lockdown [33–36].

## Strength and limitations

One big strength of this study is its robust design that encapsulates lived experiences and realities of the most vulnerable groups in concurrent humanitarian crises of Cox's Bazar. As per our knowledge, this is the first study that has laid out the most vulnerable in the context of COVID-19 and an existing humanitarian crisis. Since MVGs' lived experiences in the region throughout the COVID-19 lockdown period were comparable in terms of accessing healthcare, medicine, transportation, social support and economic conditions so the research findings are generalizable to other MVGs in the host communities adjunct to the camps and in the Rohingya camps except those with refugee status.

One limitation of this study is that it was conducted approximately two-to-three months after the lockdown period was lifted off. Such a long recall period may have introduced bias into the study. To combat this limitation, the authors source-triangulated the findings pertaining to the COVID-19 lockdown with secondary literature (newspaper articles, blogs, and Op-Ed) available from the time of lockdown.

## Conclusion

We conclude that the healthcare-seeking behavior of MVGs amid the COVID-19 pandemic in the context of Rohingya and the host communities of Cox's Bazar was influenced by several factors ranging from socioeconomic, demographic, individual, health belief-related, and institutional factors. However, these factors are not linear rather they are intertwined and their intersectionality represents diverse nuances of the lived realities of these MVGs during the COVID-19 pandemic. Based on the research findings, we recommend targeted approaches for MVGs, especially for elderly, PWDs and pregnant women to assist and encourage them for seeking formal healthcare services. For example, door-to-door services by the community health workers or trained paraprofessionals, periodic outreaches/satellite clinics at the community level by certified medical professionals in the remote locations, and involving people trusted to the community as ambassador/spokespersons in community awareness programs.

## Acknowledgments

The authors are also thankful to the larger research team members; Kazi Sameen Nasar, Saifa Raz, Abdul Jabbar Topu, ASM Nadim, Zuhrat Mahfuza Inam, and Dr. Ashrafuzzaman Khan, for their support during the data collection.

## Author Contributions

**Conceptualization:** Muhammad Riaz Hossain, Ateeb Ahmad Parray, Bachera Aktar, Sabina Faiz Rashid.

**Data curation:** Muhammad Riaz Hossain, Ateeb Ahmad Parray, Rafia Sultana.

**Formal analysis:** Muhammad Riaz Hossain, Ateeb Ahmad Parray, Rafia Sultana.

**Funding acquisition:** Sabina Faiz Rashid.

**Investigation:** Muhammad Riaz Hossain, Ateeb Ahmad Parray, Rafia Sultana, Sabina Faiz Rashid.

**Methodology:** Muhammad Riaz Hossain, Ateeb Ahmad Parray, Bachera Aktar, Sabina Faiz Rashid.

**Project administration:** Muhammad Riaz Hossain, Ateeb Ahmad Parray, Rafia Sultana, Bachera Aktar, Sabina Faiz Rashid.

**Resources:** Muhammad Riaz Hossain, Ateeb Ahmad Parray, Rafia Sultana, Bachera Aktar, Sabina Faiz Rashid.

**Software:** Muhammad Riaz Hossain, Ateeb Ahmad Parray, Rafia Sultana.

**Supervision:** Bachera Aktar, Sabina Faiz Rashid.

**Validation:** Muhammad Riaz Hossain, Ateeb Ahmad Parray, Rafia Sultana, Bachera Aktar, Sabina Faiz Rashid.

**Visualization:** Muhammad Riaz Hossain, Ateeb Ahmad Parray, Rafia Sultana, Bachera Aktar.

**Writing – original draft:** Muhammad Riaz Hossain.

**Writing – review & editing:** Muhammad Riaz Hossain, Ateeb Ahmad Parray, Rafia Sultana, Bachera Aktar, Sabina Faiz Rashid.

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
