## [Editor Report · Decision Letter 0]

20 Apr 2022

PGPH-D-22-00229

Factors influencing healthcare-seeking behavior of most vulnerable groups amid the COVID-19 pandemic in the humanitarian crisis in Cox’s Bazar, Bangladesh: findings from an exploratory qualitative study

Dear Dr. Hossain,

Thank you for submitting your manuscript to PLOS Global Public Health. After careful consideration, we feel that it has merit but does not fully meet PLOS Global Public Health’s publication criteria as it currently stands. Therefore, we invite you to submit a revised version of the manuscript that addresses the points raised during the review process.

Please submit your revised manuscript by . If you will need more time than this to complete your revisions, please reply to this message or contact the journal office at globalpubhealth@plos.org. Please include the following items when submitting your revised manuscript:

We look forward to receiving your revised manuscript.

Kind regards,

Judith McCool, PhD

Academic Editor

Journal Requirements:

1. Please state in the Methods:

- Whether the Institutional Review Board (IRB) approved use of oral consent

- How oral consent was documented

For more information, please see our guidelines for human subjects research: https://journals.plos.org/globalpublichealth/s/submission-guidelines#loc-human-subjects-research

2. Please provide the full name of the Ethics Committee or IRB in the Ethics Statement section in the manuscript file.

3. Please send a completed 'Competing Interests' statement, including any COIs declared by your co-authors. If you have no competing interests to declare, please state "The authors have declared that no competing interests exist". 

4. Please provide  separate figure files in .tif or .eps format only and remove any figures embedded in your manuscript file.  Please ensure that all files are under our size limit of 20MB.  

For more information about how to convert your figure files please see our guidelines: Once you've converted your files to .tif or .eps, please also make sure that your figures meet our format requirements

5. Please ensure that the funders and grant numbers match between the Financial Disclosure field and the Funding Information tab in your submission form.

Additional Editor Comments (if provided):

Thank you for presenting a comprehensive descriptive and analytical account of the specific challenges faced by the Rohingya populations living in Cox's Bazar during the COVID-19 pandemic. It is a compelling and important issue; the rhetoric of equity and redistribution of meagre resources being under question, alongside the specific strategies used by MVGs when such pressures are exerted. I have one major issue with the manuscript as it stands. It is by far too long to be accessible for most readers. At over 10,000 words in length is more aligned to a full dissertation than a manuscript. I am therefore requesting the authors edit the manuscript length to reduce it to around 4,500 words (although I note that there is no word limit on submissions). There are several areas where these edits could be made; the introduction, the tables could be reduced to one or two (one in the methods, perhaps) and one to describe the sample (in the results). The results are interesting and important, but again, some editing and merging of sections would only serve to improve the readability of the text. A clearer series of statements to describe the value this work adds to the broader literature, service response (background) and what this work will mean in the future (discussion) would again be of value to the reader. There is no doubt in my mind that pandemic preparedness needs to consider how to best support the most vulnerable populations; health seeking behaviors provide valuable insight into where provisions are inadequate, where pressure points are, how individual and community / collectives needs are expressed and met.

The abstract background provides an insight into the work needed on this front; that this work has not been undertaken and that the Rohingya are uniquely vulnerable is of course, an important driver for this work, but how this work contributes to broader (or even national, regional) challenges and objectives for fragile and vulnerable populations during COVID-19 would anchor this substantial piece of work.

The Figure included in the background, does this need to be presented in the paper? Could it be referred to in the text and referenced?

Results present 'partial findings' - please explain what is mean by this? Is there another paper in publication or preparation?
---

## [Decision Letter · Decision Letter 1]

27 Oct 2022

PGPH-D-22-00229R1

Exploring healthcare-seeking behavior of most vulnerable groups amid the covid-19 pandemic in the humanitarian context in Cox’s Bazar, Bangladesh: findings from an exploratory qualitative study

Dear Dr. Hossain,

Thank you for submitting your manuscript to PLOS Global Public Health. After careful consideration, we feel that it has merit but does not fully meet PLOS Global Public Health’s publication criteria as it currently stands. Therefore, we invite you to submit a revised version of the manuscript that addresses the points raised during the review process.

We look forward to receiving your revised manuscript.

Kind regards,

Judith McCool, PhD

Academic Editor

Journal Requirements:

Additional Editor Comments (if provided):

Reviewers' comments:

Reviewer's Responses to Questions

**Comments to the Author**

1. If the authors have adequately addressed your comments raised in a previous round of review and you feel that this manuscript is now acceptable for publication, you may indicate that here to bypass the “Comments to the Author” section, enter your conflict of interest statement in the “Confidential to Editor” section, and submit your "Accept" recommendation.

Reviewer #1: (No Response)

Reviewer #2: (No Response)

2. Does this manuscript meet PLOS Global Public Health’s publication criteria? Is the manuscript technically sound, and do the data support the conclusions? The manuscript must describe methodologically and ethically rigorous research with conclusions that are appropriately drawn based on the data presented.

Reviewer #1: Yes

Reviewer #2: Partly

3. Has the statistical analysis been performed appropriately and rigorously?

Reviewer #1: (No Response)

Reviewer #2: N/A

4. Have the authors made all data underlying the findings in their manuscript fully available (please refer to the Data Availability Statement at the start of the manuscript PDF file)?

Reviewer #1: Yes

Reviewer #2: No

5. Is the manuscript presented in an intelligible fashion and written in standard English?

Reviewer #1: Yes

Reviewer #2: No

6. Review Comments to the Author

Reviewer #1: Thanks to the authors for this research, which studied the healthcare-seeking behaviour of the most vulnerable groups of two different communities in Cox´s Bazaar during the COVID-19 pandemic

General impression

The research topic is important because it examines the well-being of the most vulnerable groups (refugees) in a highly challenging humanitarian context. Having female researchers for the interviews is likely to increase the reliability of responses to sensitive topics from woman participants. More studies like this are needed. Good job!

The research question is straightforward, and the method used suits the research. Ethical considerations have been taken into account. The results are mostly clear and consistent. The information obtained from the research can be used to improve infectious disease preparedness in humanitarian crises.

The language is clear and easy to read, although some sections contain minor errors or omissions.

Introduction

- Since the situation of Rohingya is not very familiar to all readers, I would suggest that a little more background for the study should be written in the introduction section (or settings). For example, where do Rohingyas come from, when did they arrive in Bangladesh and why. A little more information about the population group, such as religion, is needed. The same applies to the local host population, as there is little information about its background. It would also be good to describe the relations between these two communities. Adding such information helps the reader better understand the research results and put them in context.

- line 85: define the abbreviation WASH

Materials and Methods

- line 122: authors say that they have conducted an exhaustive literature review. However, the article does not present other studies that are related to the topic. It would be desirable to highlight a few other such studies. The same issues have been researched in other conflict settings, such as in Syria.

- line 127: Even though Andersen's model is well known, the authors could briefly describe the method.

- lines 129-130. The authors stated that they had left certain factors out of the analysis, such as religion (in case of refugees it is Islam). However, there are several points in the results that can be better explained using religious norms. Interestingly, there are multiple references to religion in the results. For example, Rohingya women are not allowed to go outside without an escort, while local women can. See, e.g. lines 212-124 However, unlike the elderly Rohingya participants, their dependency on family members was driven mainly by two factors; (a) strong socio-cultural, religious, and gender norms that restricted their mobility to venture outside without their family members… It remains unclear to the reader, what excluding these factors means in practice or why it should be done at all. I think this is a significant inconsistency in the work.

- line 129: Figure 1. is missing ….our exploration of factors influencing healthcare-seeking behavior in the Rohingya and host community contexts (Fig 1)

- The study lacks information about how the interviewees were selected for the study. At what point was it decided to stop continuing the interviews? Is the sampling reliably representative of the whole population and likely to be without biases? I think this an important consideration that should be added.

Results

- lines 222 - 224 As a result, many of them admitted seeking help 223 from their male family members whenever they faced any health-related issues or menstrual kit. Here should be “ or when women needed the mensural kit?

- lines 264- 272 I think this is background info and should be transferred to context / setting chapter

- line 344 paled:shold be placed?

- lines 456 - 457 Most of their treatment was from the paracetamol group, and doctors prescribed those without proper examination. Due to this reason, patients were not recovering as early as the pre-pandemic period. This sentence is a bit problematic. Providing paracetamol has been almost the only possible medication for symptoms of COVID-19 since there is no cure for the disease. This argument shows that the patients did not know the facts about the virus. The interviewees felt that they did not receive proper treatment (because they did not know that it did not exist). While failing to provide life-sustaining medical aid to critically ill patients would constitute a medical performance failure, giving paracetamol as the typical treatment for patients without complications is actually proper practice. Objectively sub-par treatment should be separated from the subjective experience of the patients. In itself this phenomenon could be an interesting point for further study. Patients might have appreciated the treatment better, had they been informed about COVID transmission and feasible medication.

- It was noted at several points that financial issues were one factor limiting the availability of medical services. It would be helpful to the reader if the funding structure of different healthcare services was described in more detail. For which services the patients themselves were expected to pay? Was lack of state or NGOs funds a significant factor in this sense?

Discussion and Conclusion

The findings appear to be logical and believable, consistently supporting previous findings. There are some parts of the discussion where telling apart findings of previous publications from the ones done in this study is challenging. It would be beneficial to be explicitly clear about the new core findings in this section. Also, the results could be connected to an even broader perspective by comparing them against comparable studies from contemporary conflict settings such as Syria.

It was noted that the results could not be generalized to other Rohingya camps and host communities. What is the maximum extent of reliable generalizability of the results, and what are the limiting factors? Should the article be viewed primarily as a case study?

Overall, a welcome contribution to our understanding of healthcare-seeking behaviour of the most vulnerable groups in complex humanitarian settings.

Reviewer #2: (No Response)

7. PLOS authors have the option to publish the peer review history of their article (what does this mean?). If published, this will include your full peer review and any attached files.

**Do you want your identity to be public for this peer review?** For information about this choice, including consent withdrawal, please see our Privacy Policy.

Reviewer #1: **Yes: **Agneta Kallström

Reviewer #2: No

---

## [Decision Letter · Decision Letter 2]

15 Feb 2023

PGPH-D-22-00229R2

Exploring healthcare-seeking behavior of most vulnerable groups amid the covid-19 pandemic in the humanitarian context in Cox’s Bazar, Bangladesh: findings from an exploratory qualitative study

Dear Dr. Muhammad Riaz Hossain,

Thank you for submitting your manuscript to PLOS Global Public Health. After careful consideration, we feel that it has merit but does not fully meet PLOS Global Public Health’s publication criteria as it currently stands. Therefore, we invite you to submit a revised version of the manuscript that addresses **the point 6 of the Reviewer 3** as detailed below. 

We look forward to receiving your revised manuscript.

Kind regards,

Thu-Anh Nguyen

Academic Editor

Journal Requirements:

Reviewers' comments:

Reviewer's Responses to Questions

**Comments to the Author**

1. If the authors have adequately addressed your comments raised in a previous round of review and you feel that this manuscript is now acceptable for publication, you may indicate that here to bypass the “Comments to the Author” section, enter your conflict of interest statement in the “Confidential to Editor” section, and submit your "Accept" recommendation.

Reviewer #1: All comments have been addressed

Reviewer #3: All comments have been addressed

2. Does this manuscript meet PLOS Global Public Health’s publication criteria? Is the manuscript technically sound, and do the data support the conclusions? The manuscript must describe methodologically and ethically rigorous research with conclusions that are appropriately drawn based on the data presented.

Reviewer #1: Yes

Reviewer #3: Yes

3. Has the statistical analysis been performed appropriately and rigorously?

Reviewer #1: Yes

Reviewer #3: N/A

4. Have the authors made all data underlying the findings in their manuscript fully available (please refer to the Data Availability Statement at the start of the manuscript PDF file)?

Reviewer #1: Yes

Reviewer #3: No

5. Is the manuscript presented in an intelligible fashion and written in standard English?

Reviewer #1: (No Response)

Reviewer #3: Yes

6. Review Comments to the Author

Reviewer #1: The article has been significantly clarified and I think it is ready for publication. The topic of the work is significant because it provides important additional information about marginalized groups of people and their health care.

**Reviewer #3: **The authors have addressed previous reviewer comments satisfactorily.

There is however an issue with the data availability statement - the authors state that all data is presented in the manuscript, which seems very unlikely given they interviewed 48 people. The authors should use the guidelines for qualitative data here: https://journals.plos.org/globalpublichealth/s/data-availability to complete their data availability statement. As it's common that qualitative data is subject to sharing restrictions given the identifiable and often sensitive nature of the data, the authors can use this statement to explain any restrictions on data sharing (identifiability, data sharing not covered by participants' consent).

7. PLOS authors have the option to publish the peer review history of their article (what does this mean?). If published, this will include your full peer review and any attached files.

**Do you want your identity to be public for this peer review?** For information about this choice, including consent withdrawal, please see our Privacy Policy.

Reviewer #1: **Yes: **Agneta Kallström

Reviewer #3: No

---

## [Editor Report · Decision Letter 3]

27 Feb 2023

Exploring healthcare-seeking behavior of most vulnerable groups amid the covid-19 pandemic in the humanitarian context in Cox’s Bazar, Bangladesh: findings from an exploratory qualitative study

PGPH-D-22-00229R3

Dear Dr Hossain,

We are pleased to inform you that your manuscript 'Exploring healthcare-seeking behavior of most vulnerable groups amid the covid-19 pandemic in the humanitarian context in Cox’s Bazar, Bangladesh: findings from an exploratory qualitative study' has been provisionally accepted for publication in PLOS Global Public Health.

Best regards,

Thu-Anh Nguyen

Academic Editor